# Phthalates exposure and serum uric acid level in patients with Crohn's disease: A cross-sectional study

Xinghuang Liu [1], De Qingzhuoga[1], Danping Xiong[2], Liang Wang[3], Xiaohua Hou[1], Bai Tao[1], Liangle Yang[4], Liangru Zhu[1]*, Lei Tu[1]*

1 Division of Gastroenterology, Union Hospital, Tongji Medical College, Huazhong University of Science and Technology, Wuhan, PR China, 2 Division of Gastroenterology, Puren Hospital, Wuhan, PR China, 3 Department of Optics, Research Institute of HUST in Shenzhen, Huazhong University of Science and Technology, Shenzhen, PR China, 4 Department of Occupational and Environmental Health, School of Public Health, Tongji Medical College, Huazhong University of Science and Technology, Wuhan, PR China

☯ These authors contributed equally to this work.
* zhuliangru@hust.edu.cn (LZ); tulei_1985@126.com (LT)

## Abstract

### Objective

Crohn's disease (CD) is closely associated with disorders of uric acid metabolism. Our previous research found an association between phthalate exposure and oxidative stress in CD, suggesting a potential role for phthalates in metabolic disorders. Therefore, this study aims to examine their influence on uric acid metabolism in patients with CD.

### Methods

We designed a cross-sectional study involving 117 patients with CD. Ten urinary phthalates metabolites (mPAEs) were detected by gas chromatography-tandem mass spectrometry, and the serum uric acid (SUA) levels were tested. Correlation analysis and Bayesian kernel machine regression (BKMR) models were applied separately to evaluate the associations.

### Results

The prevalence of hyperuricemia was 12.8% (15/117) in CD patients. None of them were obese or had abnormal renal function. In males, we identified significant positive associations between SUA and eight mPAEs (MMP, MIBP, MBP, MBzP, MOP, MEOHP, MEHHP, & MECPP). However, no positive associations between mPAEs and SUA were found in females. After BKMR analysis and multivariate adjustments, we found that the average SUA (μmol/L) increased by 1.36-fold, and the odds ratio for hyperuricemia increased by 1.25-fold, when overall phthalates exposure increased from 25% to 75% in male CD patients. This suggests a potential link

**Data availability statement:** The raw data generated in this study have been deposited in the Mendeley Data repository (https://doi.org/10.17632/rygn7ywndg.1). The data are publicly available. Due to ethical restrictions aimed at protecting patient confidentiality, direct identifiers have been removed from the dataset. The dataset contains the information of Crohn's disease patients used in the research, including demographics, laboratory results, and environmental pollution exposure biomarkers. For requests concerning more detailed data or other supporting materials, interested researchers should contact Xinghuang Liu via email (xinghuang-liu@qq.com), or Lei Tu via email (tulei_1985@126.com). De-identified data are also available upon request from the Ethics Committee of Wuhan Union Hospital via email (ethics@wuhunion.com) for researchers who meet the criteria for access to confidential data.

**Funding:** This work was supported by Natural Science Foundation of China (82370543) (LT); Key Research and Development Program of Hubei Province (2024BCB016) (LT), Natural Science Foundation of Wuhan (2024040801020351) (LT) Interdisciplinary Research Program of HUST (2024JCYJ039) (LT&LW); Science Research Foundation of Union Hospital (2022xhyn009) (LT); Teaching Reform Project of the First Clinical College (202120) (LT) and Natural Science Foundation of Hubei Province (2023AFB807) (TB).

**Competing interests:** The authors have declared that no competing interests exist.

between phthalates exposure and uric acid metabolism in male patients with CD. Furthermore, oxidative stress mediated approximately 5% of the association, indicating it is a partial, but not primary, mechanism in this process.

## Conclusions

Phthalates exposure positively correlated with SUA in male CD patients. Effective PAE exposure control in patients with CD may reduce the risk of hyperuricemia.

---

## Introduction

Hyperuricemia (HUA) has become a worldwide public health issue [1]. The recent study reports a significant upward trend in the prevalence of hyperuricemia, with an estimated prevalence rate of 14.0% [2]. Hyperuricemia is a significant risk factor for the development of gout [3]. What's more, elevated serum uric acid (SUA) is linked to increased risks of chronic kidney disease, cardiovascular events, longer hospitalization, and death [4–6].

The intestine is an important organ for uric acid metabolism. Previous physiological studies have found that approximately one-third of uric acid is excreted in the intestine, with the remaining two-thirds excreted in urine [7]. When kidney function is impaired, the intestine can compensate by increasing uric acid excretion [8]. Further research found that the intestinal tract is a crucial organ for both the production and absorption of uric acid [9]. Thus, impaired intestinal function leads to uric acid accumulation, causing hyperuricemia [10]. Increased intestinal permeability in certain intestinal diseases allows the entry of intestinal bacterial products into the body, triggering an inflammatory response that affects uric acid metabolism [11]. The intestinal microbiota participate in uric acid metabolism, and when the microbial community is imbalanced, it may increase uric acid levels [9,12].

Crohn's disease (CD), an inflammatory bowel disease (IBD), is a chronic, progressive condition with relapsing and remitting features, characterized by inflammation of the gastrointestinal tract [13]. Important pathophysiologic mechanisms of CD include impaired intestinal barrier function and disturbances in intestinal microbiota [14]. In a large, population-based sample, the prevalence of gout increased significantly among patients with CD [15]. Given that obesity is a significant risk factor for high serum uric acid levels, hyperuricemia, and gout, and that patients with CD typically have a lower body mass index (BMI) than the general population, it is likely that the way CD patients metabolize uric acid differs significantly from the general population [16–18]. So far, research suggests that hyperuricemia in IBD has been linked to some pathogenetic mechanisms, like immune, microbial, and genetic factors [19]. However, the essential role of environmental pollution factors as the etiology in the elevation of SUA in CD patients has never been recognized.

Phthalates (PAEs) are ubiquitous environmental contaminants widely used in industrial and consumer products as plasticizers or additives [20]. Because of the absence of chemical bonding in polymers, phthalates are quickly released from

plastics into the environment, causing widespread exposure among the general population via food ingestion, skin absorption, and air inhalation [21]. In China, human exposure to PAEs has increased during the last decade (e.g., MMP, 8.7% increase per year; MnBP, 34.4% increase per year) [22].

Our previous research found that PAEs exposure was closely related to CD activity, and the association could be mediated by oxidative stress, which indicated the potential role of phthalates in metabolic disorders and emphasized the pivotal role of oxidative stress in their mechanism [23]. Several studies have identi, i.e.,d a positive relationship between PAEs and uric acid levels in the general population [24,25]. Previous studies have also indicated a paradoxical association between uric acid and the oxidative stress process [26,27]. Therefore, the current study aimed to examine the association between phthalates exposure and serum uric acid in Crohn's disease patients, and the role of oxidative stress in this process was also explored.

## Materials and methods

### Study population

This cross-sectional study was conducted at Wuhan Union Hospital (Wuhan, China) between 01 August 2021 and 30 October 2021. Hospitalized patients diagnosed with Crohn's disease (CD), according to the ECCO-ESGAR Guideline, were recruited [28]. Exclusion criteria: (1) history of surgical procedures or enterostomy; (2) significant comorbidities (e.g., infectious diseases, asthma, allergies, chronic kidney disease and autoimmune thyroid diseases), or acute diseases (e.g., food poisoning); (3) concomitant enteric pathogen infection (e.g., Clostridium difficile, Cytomegalovirus, Epstein-Barr virus, etc.); (4) age lower than 16 years old. Patients with repeated hospital admissions were included only once. Informed consent was obtained from all participants. To avoid selection bias, we implemented a total population approach wherein every CD patient admitted to the ward during the study window was considered without differentiation. The study complied with the Helsinki Declaration and was approved by the Ethics Committee of Wuhan Union Hospital (No.S435).

### Data and sample collection

General characteristics, including age, sex, height, weight, comorbidities, symptoms, and medical and surgical history, were collected. All participants underwent symptom assessment to calculate Crohn's disease Harvey-Bradshaw index (HBI) [29], which included general well-being, stool frequency, abdominal pain, complications of CD, and presence of an abdominal mass. Serum uric acid, serum creatinine, and estimated glomerular filtration rate (eGFR) were measured by the blood samples collected the next morning after admission. Meanwhile, all participants were asked to provide first-morning urine samples (from 6:30 a.m. to 8:00 a.m.). We used the cutoff point for uric acid of 416 μmol/L for men and 357 μmol/L for women to diagnose hyperuricemia [30,31]. Abnormal renal function was defined as the eGFR less than 90 ml·min$^{-1}$/1.73 m$^2$ [32].

### Measurement of phthalates and oxidative stress biomarkers

It is difficult to directly measure the concentration of phthalates (PAEs) or the ubiquity of PAEs in blood. Phthalate metabolites (mPAEs) are widely used for quantifying human exposure to phthalates in urine samples due to their noninvasiveness, high detection rates, and ease of collection [33,34].

The concentrations of ten urinary mPAEs, including monomethyl phthalate (MMP), monoethyl phthalate (MEP), mono-iso-butyl phthalate (MiBP), mono-n-butyl phthalate (MBP), mono-(2-ethylhexyl) phthalate (MEHP), mono-benzyl phthalate (MBzP), mono-n-octyl phthalate (MOP), mono-2-ethyl-5- oxohexyl phthalate (MEOHP), mono-2-ethyl-5-hydroxyhexyl phthalate, (MEHHP), and mono-(2-ethyl-5-carboxypentyl) phthalate (MECPP) were measured using a gas chromatography-tandem mass spectrometry (GC-MS/MS) using Agilent 8890 gas chromatography system coupled with Agilent 7010B triple quadrupole mass spectrometer (Agilent, Palo Alto, CA, USA) based on a reliable approach outlined

previously described [35–37]. More detailed information on methods and detection quality is provided in the S1 File and in S1 and S2 Tables.

The BS-200 automatic biochemistry analyzer (Shenzhen Mindray Bio-Medical Electronics Co., Ltd.) was used to determine urinary creatinine concentrations. In this study, mPAEs were adjusted for the serum-to-urine creatinine ratio and reported as ng/L.

PAE metabolites were classified into two groups based on molecular weights of their parent compounds: including low molecular weight (LMW) mPAEs (MMP, MEP, MiBP, MBP, MBzP) and high molecular weight (HMW) mPAEs (MEHP, MOP, MEOHP, MEHHP, MECPP) [38]. DEHP metabolites (MEHP, MEOHP, MEHHP, MECPP; mDEHP) and total mPAEs (sum of 10 individual mPAEs) were also analyzed [39].

Concentrations of oxidative stress biomarkers (8-OHdG and 8-iso-PGF2α) in urine samples were evaluated with enzyme-linked immunosorbent assay (ELISA) kits (Shanghai Shunshi, China) following the manufacturer's instructions. Concentrations of urinary 8-OHdG and 8-iso-PGF2α were adjusted by the ratio of serum creatinine to urine creatinine.

### Statistical analysis

Considering the significant differences in uric acid metabolism between men and women [40,41], this study discussed each sex separately. Continuous variables were reported as median (IQR), and categorical variables as frequencies (percentages). To achieve normality, serum uric acid (SUA) levels were natural-log (ln) transformed.

The association between mPAEs and SUA levels or hyperuricemia was evaluated using the following approach: **1)Correlation and Regression:** Spearman correlation was used for bivariate analysis. Multivariate linear and logistic regression models, adjusted for age, BMI, and HBI, were employed to estimate standardized coefficients (β) and odds ratios (ORs) with 95% confidence intervals (CIs) calculated via the bootstrap method. The interaction between mPAEs and disease activity (HBI) was explored using stepwise multivariate linear regression by adding interaction terms (mPAEs × HBI) to the model. **2)Joint Exposure Analysis:** To account for the multi-pollutant nature and high correlation of mPAEs, Bayesian kernel machine regression (BKMR) was implemented. This model estimated the joint effects of the mixture on SUA and hyperuricemia while identifying major contributors through hierarchical variable selection [42]. The BKMR model utilized a Markov chain Monte Carlo (MCMC) algorithm with 50,000 iterations. Overall associations were estimated by setting the mixture at specific percentiles (25th-75th) relative to the 50th percentile. **3)Mediation Analysis:** We used a mediation model to explore whether oxidative stress (M) mediated the relationship between mPAEs (X) and SUA levels (Y). Robust standard errors and bootstrapping were applied to handle non-normal distributions.

Standardized collection protocols ensured a complete dataset for all 117 participants with zero missing values in core variables. All analyses were conducted using R software (version 4.3.3) with a two-tailed significance threshold of $P < 0.05$.

## Results

A total of 163 CD patients were interviewed, of whom 117 (71.8%) were included in the analysis. The flowchart of the study procedure is shown in Fig 1.

### Demographic, clinical characteristics, and phthalates exposure of the study population

The demographic, clinical characteristics, and phthalates exposure of CD patients were summarized in Table 1. All the patients had normal renal function (eGFR > 90 ml·min$^{-1}$/1.73 m²) and were not obese (BMI > 30 kg/m²). The prevalence of hyperuricemia was 12.8% (15/117) in CD patients, similar to that of the general population [2]. Despite significant differences in demographic and SUA levels between male and female patients, there were no significant differences in the proportion of hyperuricemia and mPAEs.

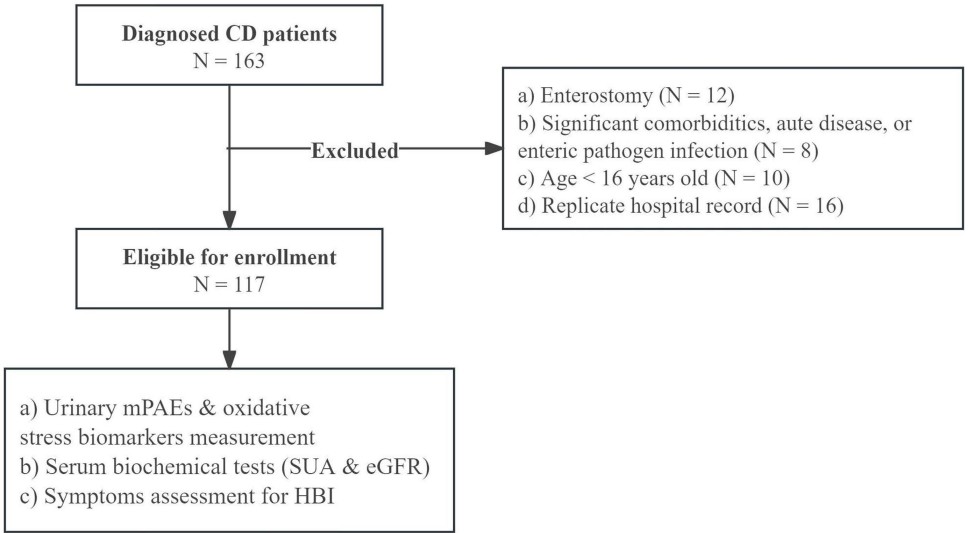

**Fig 1. Study flowchart.**

## Associations between SUA level and mPAEs

Both female and male patients exhibited significant internal correlations among the various mPAEs (Fig 2). In males, most types of mPAEs were significantly positively associated with SUA, hyperuricemia, and HBI. However, none of the mPAEs was significantly associated with SUA or hyperuricemia in females.

Multivariate linear and logistic regression models adjusted for age, BMI, and HBI were adopted to analyze the relationship between mPAEs and SUA level and hyperuricemia (Table 2). Similar to the correlation analysis results, in male patients, most types of mPAEs were still significantly associated with SUA level.

Stepwise regression analysis assessed the interaction between mPAEs and HBI, with negative results (S3 Table).

## Joint effect of mPAEs on SUA level

Because neither SUA nor mPAEs were significantly associated with eGFR or biological agents use (Fig 2), only age, BMI, and HBI were included as covariates in subsequent BKMR analyses. Positive overall associations of mPAEs mixtures with SUA, when ten urinary phthalate metabolites were concurrently at a particular percentile, ranging from 25th to 75th percentile, compared to the mixture at medians in male patients, but not female patients (Fig 3). The average SUA concentration would increase by 1.36-fold, and the odds ratio of hyperuricemia would increase by 1.25-fold when the overall phthalates exposure increased from 25% to 75% in male CD patients (S4 Table). Among the 10 phthalates, MEOHP showed the strongest association with SUA levels and hyperuricemia, with the highest posterior inclusion probabilities (PIPs) in the models (S5 Table). However, we found that none of the individual mPAEs can significantly influence the SUA levels or hyperuricemia (Fig 3).

## Mediation effect of oxidative stress on the associations between mPAEs and SUA level

As shown in Fig 2, there is a significant correlation between oxidative stress markers (8-OHdG as the biomarker of oxidative DNA damage or 8-iso-PGF2α as the biomarker of lipid peroxidation) and SUA level, as well as mPAEs in male patients [43,44]. The mediation effect model was adopted to analyze the role of oxidative stress in mediating the association between mPAEs and SUA level. In this model, SUA level was considered as the dependent variable (Y), each of

**Table 1. Demographic, clinical characteristics, and phthalate metabolites of patients with Crohn's disease.**

| Variables | Male (N = 88) | Female (N = 29) |
|---|---|---|
| Age (years) | 27.0 (20.3-33.0) | 35.0 (22.5-49.0) |
| BMI (kg/m²) | 19.2 (17.3-21.3) | 17.9 (16.1-19.5) |
| HBI | 2.0 (1.0-5.0) | 3.0 (1.0-5.5) |
| SUA (µmol/L) | 332.3 (268.8-412.6) | 241.4 (206.6-296.0) |
| Hyperuricemia (yes) | 12 (13.6%) | 3 (10.3%) |
| Biological agents use (yes) | 53 (60.2%) | 15 (51.7%) |
| eGFR (ml·min⁻¹/1.73 m²) | 122.2 (112.6-131.7) | 118.9 (108.1-131.3) |
| 8-OHdG (ng/L) | 1.4 (1.0-2.5) | 1.6 (1.2-2.9) |
| 8-iso-PGF2α (ng/L) | 0.9 (0.5-1.5) | 1.3 (0.7-2.2) |
| mPAEs (ng/L) | | |
| MMP | 20.1 (10.2-43.3) | 11.7 (5.7-35.7) |
| MEP | 102.0 (57.9-173.5) | 86.5 (42.4-163.1) |
| MIBP | 138.2 (83.3-233.1) | 111.4 (72.9-175.7) |
| MBP | 915.2 (504.9-1566.6) | 645.0 (424.3-1425.8) |
| MEHP | 26.0 (6.3-75.9) | 26.2 (4.7-63.3) |
| MBzP | 6.8 (4.0-10.8) | 7.6 (5.4-11.7) |
| MOP | 2.3 (1.0-5.4) | 1.5 (0.9-5.9) |
| MEOHP | 56.1 (33.3-110.8) | 65.3 (39.0-116.3) |
| MEHHP | 81.0 (32.9-202.8) | 85.3 (35.9-188.8) |
| MECPP | 134.0 (73.2-266.2) | 123.2 (70.3-177.3) |
| Total mPAEs | 1832.0 (1109.8-2945.0) | 1371.1 (983.1-2762.1) |
| Types of mPAEs (ng/L) | | |
| LMW mPAEs | 1250.0 (756.6-2135.3) | 913.1 (655.9-1987.8) |
| HMW mPAEs | 361.9 (222.8-672.0) | 364.3 (212.5-594.8) |
| mDEHP | 352.1 (214.5-638.0) | 354.4 (196.2-582.4) |

Data are median (IQR) or n (%). mPAEs, PAE metabolites; Total mPAEs: sum of ten individual mPAEs; LMW mPAEs, low molecular weight mPAEs; HMW mPAEs, high molecular weight mPAEs; mDEHP, DEHP metabolites.

mPAEs as the independent variable (X), 8-OHdG or 8-iso-PGF2α as the mediator variables (M), and age, BMI & HBI were included as confounding variables.

Mediation of oxidative stress was present. The compound 8-OHdG significantly mediated the effects of four mPAEs (MMP, MIBP, MOP & MECPP) on uric acid, with the proportion mediated ranging from 13.84% to 49.48% (S6 Table). Additionally, 8-iso-PGF2α showed similar results (S7 Table). The oxidative stress, assessed by two distinct biomarkers, played a partial mediating role: approximately 5.56% for 8-OHdG and 4.62% for 8-iso-PGF2α, indicating that oxidative stress is a minor but consistent pathway linking phthalate exposure to elevated SUA levels.

## Discussion

In this cross-sectional study of patients with CD, we found a positive association between phthalate exposure and SUA levels, particularly among male patients. BKMR analysis of phthalates mixtures revealed that when overall exposure increased from the 25th to the 75th percentile, the average SUA concentration increased by 1.36-fold, and the odds of hyperuricemia increased by 1.25-fold, after adjustment for age, BMI, and disease activity (HBI). Furthermore, mediation analysis indicated that oxidative stress partially contributed to this association.

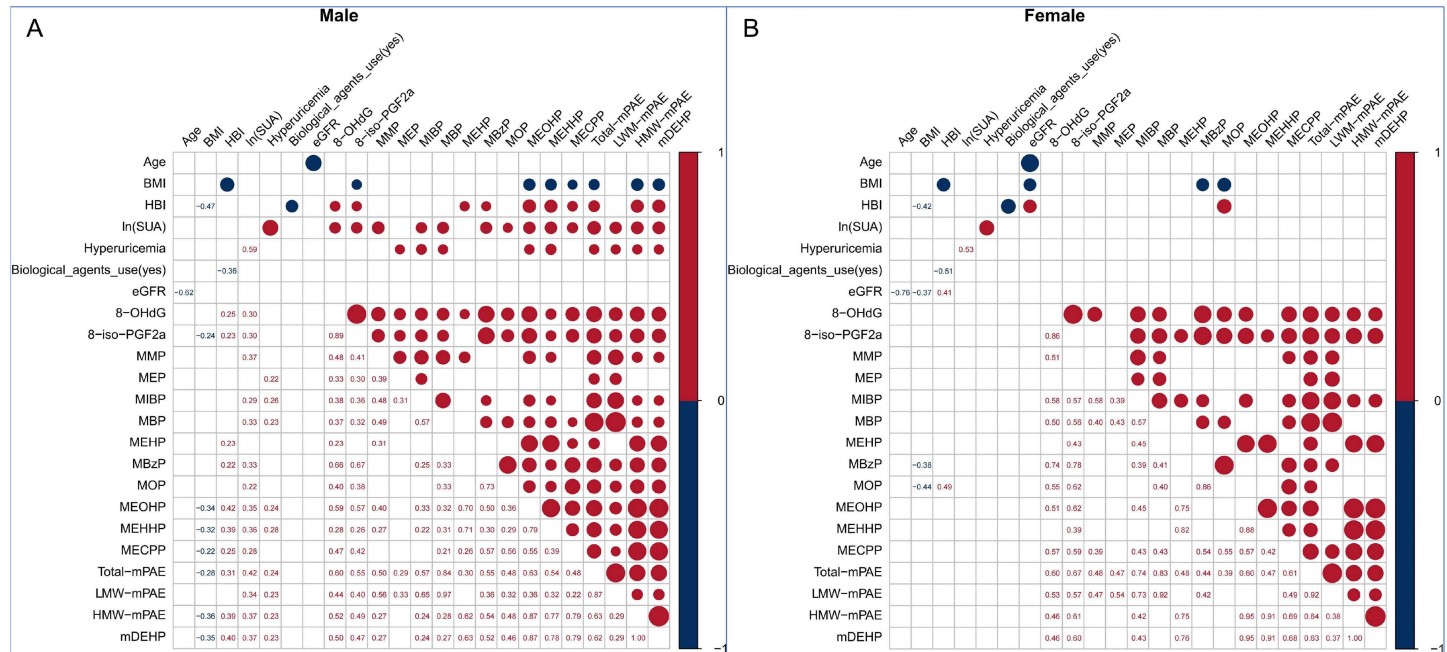

**Fig 2. Correlation Heatmaps between the SUA Level and mPAEs.** The number showed the correlation coefficients, with the dots' size matching the coefficients' absolute value. Only significant correlations were displayed..

**Table 2. Multivariate regression for the relationship between mPAEs and SUA or hyperuricemia.**

| mPAEs | Male (N = 88) | | Female (N = 29) | |
|---|---|---|---|---|
| | SUA level (In SUA) β (95%CI) | Hyperuricemia OR (95%CI) | SUA level (In SUA) β (95%CI) | Hyperuricemia OR (95%CI) |
| MMP | 0.34 (0.18, 0.47)* | 1.66 (0.92, 3.10) | 0.20 (−0.89, 0.79) | 3.59 (1.16, 79.70) |
| MEP | 0.16 (0.05, 0.76) | 1.17 (0.59, 1.95) | 0.11 (−0.60, 0.48) | 2.38 (0.84, 9.23) |
| MIBP | 0.20 (0.02, 0.52) | 2.32 (1.17, 6.71) | 0.05 (−0.45, 0.62) | 2.00 (0.61, 8.33) |
| MBP | 0.34 (0.16, 0.55)* | 2.17 (1.22, 4.28)* | 0.37 (−0.30, 0.66) | 4.12 (0.78, 52.37) |
| MEHP | 0.13 (−0.04, 0.50) | 1.29 (0.70, 2.21) | 0.31 (−0.39, 0.51) | 2.45 (0.90, 8.27) |
| MBzP | 0.30 (0.13, 0.55)* | 1.30 (0.68, 2.28) | 0.18 (−0.41, 0.55) | 3.11 (0.93, 18.21) |
| MOP | 0.22 (0.06, 0.47)* | 1.11 (0.54, 1.94) | 0.07 (−0.52, 0.68) | 2.72 (0.51, 16.91) |
| MEOHP | 0.38 (0.20, 1.09)* | 2.23 (1.23, 5.17)* | 0.34 (0.08, 0.68) | 1.80 (0.48, 7.63) |
| MEHHP | 0.39 (0.22, 0.90)* | 2.31 (1.26, 5.15)* | 0.39 (0.10, 0.71) | 2.98 (0.89, 20.59) |
| MECPP | 0.17 (−0.04, 0.64) | 1.24 (0.64, 2.11) | 0.31 (−0.16, 0.77) | 7.31 (1.39, 745.37) |
| Total mPAEs | 0.46 (0.25, 0.84)* | 2.99 (1.43, 8.23)* | 0.38 (−0.14, 0.62) | 3.55 (0.99, 34.31) |
| LMW mPAEs | 0.36 (0.18, 0.63)* | 2.24 (1.22, 4.85)* | 0.34 (−0.28, 0.60) | 3.75 (0.90, 39.58) |
| HMW mPAEs | 0.36 (0.18, 0.74)* | 1.93 (1.08, 3.70)* | 0.38 (0.07, 0.62) | 3.28 (1.01, 21.39) |
| mDEHP | 0.35 (0.18, 0.76)* | 1.93 (1.08, 3.71)* | 0.38 (0.05, 0.61) | 3.28 (1.01, 21.61) |

Linear regression for SUA and logistic regression for hyperuricemia. The models were adjusted for age, BMI, and HBI. Abbreviation: β, the standardized estimate coefficient; OR, standardized odds ratio; CI, confidence interval. * P-value < 0.05.

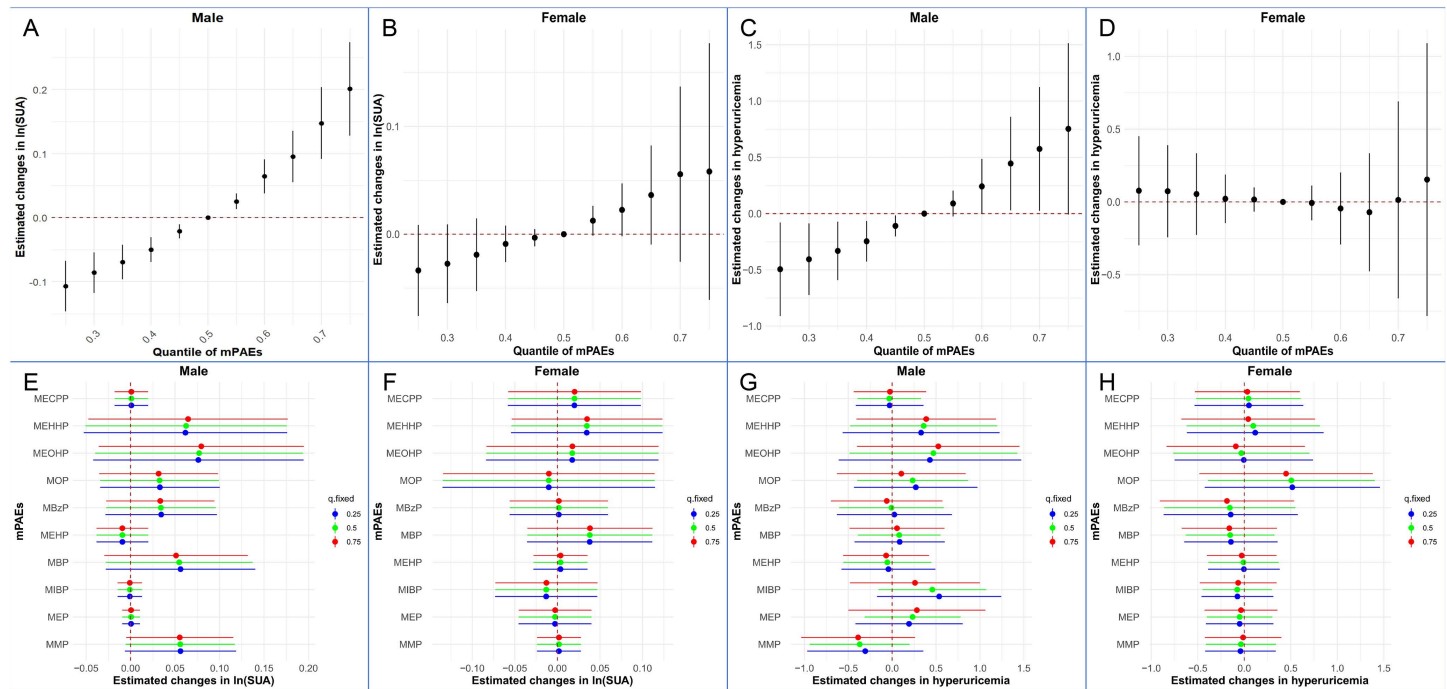

**Fig 3. Joint and Individual Effects of Phthalate Metabolite (mPAEs) Mixtures on Serum Uric Acid (SUA) and Hyperuricemia Risk Stratified by Sex.** Panels A–D: Overall Mixture Effects. These panels illustrate the joint effect of the mPAE mixture on ln-transformed SUA levels (A for males, B for females) and the risk of hyperuricemia (C for males, D for females). The plots display the estimated change in the outcome (with 95% Confidence Intervals) when all metabolites in the mixture are fixed at a specific quantile compared to when they are all at their 50th percentile. Panels E–H: Univariate Predictor-Response Relationships. These panels show the individual contribution of each phthalate metabolite to the outcome while holding all other metabolites fixed at the 25th (blue), 50th (green), or 75th (red) percentiles. Panels E & F show the estimated changes in ln(SUA) for males and females, respectively. Panels G & H show the estimated changes in hyperuricemia risk for males and females, respectively. The models were adjusted for age, BMI and HBI.

Evidence on the association between phthalates exposure and hyperuricemia or SUA level in CD patients is minimal. The metabolomics study by Tsai et al. indicated that children who consumed food contaminated with DEHP had higher SUA levels than those who consumed non-contaminated food [45]. A study in the general U.S. population from NHANES demonstrated a significant association between phthalate metabolites and elevated SUA levels, as well as an increased risk of hyperuricemia [46]. However, another study found that SUA levels were not significantly associated with phthalates in a general population sample of 1000 [47]. Evidence regarding how phthalates affect SUA levels remains inconsistent across comparable populations. Our results suggested that this relationship between phthalate exposure and SUA level remains in male patients with inflammatory bowel diseases. Given that CD is significantly positively associated with gout [15], the effect of phthalates on SUA may be complexly modulated by CD.

Previous studies have suggested that phthalates may impact SUA levels through glomerular damage. Male mice treated with DEHP (500 mg/kg or 1000 mg/kg) by gavage for 28 days exhibited glomerular atrophy [48]. Another acute toxicity animal study showed that SUA levels were significantly higher in male mice exposed to high doses of butyl benzyl phthalate (BBP) (1250 µg/kg to 5000 µg/kg) compared to controls, while animals showed significant renal and hepatic impairment [49]. However, the concentrations used in animal studies were much higher than those in the general population. All participants in our study had normal renal function, and their uric acid concentrations remained positively correlated with phthalates. Thus, phthalates may directly affect uric acid production.

Our study found that the SUA-elevating effect of phthalates may be sex-dependent. Many previous studies have made similar findings. A study found the inverse relationship between MEHP and body parameters among females but not in males [50]. Another study suggested that DEHP is associated with free thyroxine in girls, whereas urinary dibutyl phthalate (MiBP & MBP) is associated with free triiodothyronine in boys [51]. Additionally, a study on intrauterine growth restriction (IUGR) found that prenatal exposure to phthalates was associated with an increased risk of IUGR, and that male newborns were more sensitive to phthalates than females [52]. The relationship between gender and uric acid levels is significant: current diagnostic thresholds for hyperuricemia differ between men and women. A study found that estrogen may regulate uric acid levels by modulating uric acid metabolism and excretion [53]. We speculated that estrogen may also influence the relationship between phthalates and uric acid. However, due to the small number of female participants included and the lack of further exploration, such as measurement of estrogen levels, the influence of gender on the relationship between phthalates and uric acid remains unclear.

As mentioned earlier, the relationship between oxidative stress and SUA levels may be complex [26,27]. Previous studies have found a correlation between oxidative stress and uric acid metabolism [54,55]. The mediation role of oxidative stress has been reported in other PAEs-associated diseases, such as the toxicity of PAEs on the male reproductive system and preterm birth [56,57]. Moreover, our previous study showed that oxidative stress mediates the association between phthalates and CD activity [23]. This led us to speculate that oxidative stress might be an important mediator of the biological effects of phthalates. However, the mediating effect of oxidative stress, though identified, was very weak.

The limitations of our study should be acknowledged. First, we could not avoid selection bias due to the cross-sectional study design. In addition, the concentrations of mPAEs were measured in single samples of first-morning urine, which may not represent the mean body burden due to the short half-lives of these chemicals [58]. We cannot rule out the possibility of other unmeasured substances with similar exposure, which may bias the findings. The sample size of our study may have been inadequate, as only 15 participants met the diagnostic criteria for hyperuricemia. Besides, diet is a significant contributor to uric acid levels. However, it was excluded from the study due to challenges in quantification and analysis. Furthermore, it should be noted that all participants in this study presented with normal renal function and non-obese BMIs. While this minimized inter-subject variability, it may simultaneously limit the generalizability of our findings to populations with obesity or renal insufficiency.

In conclusion, our study revealed a significant correlation between phthalate exposure and serum uric acid levels in male patients with Crohn's disease, independent of the glomerular filtration rate. However, given the sample size, the results for female CD patients may require further investigation. Effective PAE exposure control in patients with CD may help reduce the risk of hyperuricemia.

## Supporting information

**S1 Table. Settings for Multiple Reaction Monitoring (MRM) Mode.**
(DOCX)

**S2 Table. The Quality Control of Urinary PAE Metabolites Qualification.**
(DOCX)

**S3 Table. Stepwise Multivariate Linear Regression for the Interaction of mPAEs and HBI for SUA.**
(DOCX)

**S4 Table. The Overall Associations of the Mixture of Ten Phthalate Metabolites with SUA level and Hyperuricemia in Male CD patients.**
(DOCX)

**S5 Table. The posterior inclusion probabilities in the Bayesian kernel machine regression model in male CD patients.**
(DOCX)

**S6 Table. Mediating effects of 8-OHdG on the association of mPAEs and SUA level in male CD patients.**
(DOCX)

**S7 Table. Mediating Effects of 8-iso-PGF2α on the Association of mPAEs and SUA Level in Male CD Patients.**
(DOCX)

**S1 File. Detial Method for the Determination of Urinary mPAEs.**
(DOCX)

## Acknowledgments

We want to extend our sincere gratitude to all the participants who selflessly participated in this clinical research study. Their invaluable contribution and cooperation have played a vital role in advancing medical knowledge and developing improved treatments. This study would not have been possible without their willingness to be involved. Thank you for your dedication and support.

## Author contributions

**Conceptualization:** Liang Wang, Liangru Zhu.

**Data curation:** De Qingzhuoga.

**Formal analysis:** Xinghuang Liu.

**Funding acquisition:** Liang Wang, Lei Tu.

**Investigation:** Danping Xiong.

**Methodology:** Xinghuang Liu.

**Project administration:** Liangle Yang.

**Resources:** Danping Xiong.

**Software:** Xinghuang Liu.

**Supervision:** Xiaohua Hou.

**Validation:** De Qingzhuoga.

**Visualization:** Xinghuang Liu.

**Writing – original draft:** Xinghuang Liu, De Qingzhuoga, Danping Xiong, Liangle Yang.

**Writing – review & editing:** Xiaohua Hou, Bai Tao, Liangru Zhu, Lei Tu.

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
