## [Decision Letter · Decision Letter 0]

15 Dec 2025

Dear Dr. Liu,

Thank you for submitting your manuscript to PLOS ONE. After careful consideration, we feel that it has merit but does not fully meet PLOS ONE’s publication criteria as it currently stands. Therefore, we invite you to submit a revised version of the manuscript that addresses the points raised during the review process.

**ACADEMIC EDITOR:**

We look forward to receiving your revised manuscript.

Kind regards,

Saheed Abiola Raheem, PhD

Academic Editor

PLOS One

Additional Editor Comments

We commend efforts and steadfastness while waiting for the reviewers’ comments. Based on the review carried out by the reviewers , I hereby suggest the major revision of the manuscript. The authors should address the concerns suggested by the reviewers accordingly and then re-submit for consideration.

Reviewers' comments:

Reviewer's Responses to Questions

**Comments to the Author**

1. Is the manuscript technically sound, and do the data support the conclusions?

Reviewer #1: Partly

Reviewer #2: Yes

2. Has the statistical analysis been performed appropriately and rigorously?

Reviewer #1: Yes

Reviewer #2: Yes

3. Have the authors made all data underlying the findings in their manuscript fully available?

Reviewer #1: No

Reviewer #2: No

4. Is the manuscript presented in an intelligible fashion and written in standard English?

Reviewer #1: No

Reviewer #2: Yes

Reviewer #1: This study investigates the association between the level of various urinary phthalates metabolites among patients with Crohn’s disease.

My major concerns as follows.

With regards to the experimental design, there is a strong imbalance between male and female patients. Furthermore, the numbers of patients with hyperuricemia are too small, which limit the detection power of the method. Among the statistical methods, only the association analysis was applied, including Spearman correction, the generalized linear models. There is no comparison between those patients with hyperuricemia and those without it. Additionally, they use 10 urinary phthalates metabolites as a proxy for the phthalates exposure, which could be distorted by the metabolite functions of the patients. Why did not measure the levels in the serum since it has been used for uric acid measurement?

Minor points.

(1) The tables and figures presented the same results. There are too many redundant data.

(2) The writings should be improved. There are many expression ambiguity and errors. For example, Lines 248-249 include redundant words and letters. Lines 103-104 include two acronyms for mPAEs while the first one seems an awkward appearance. Line 218, the statement “MEOHP was identified to be the major contributor to SUA levels” is too strong. The true meaning is the strongest association. The data did not lead to the conclusion of ‘major contributor’. “P-vale” in many places. The few sentences in the first paragraph of Discussion repeated themself.

Reviewer #2: Thank you for giving me the opportunity to review this paper. The author had conducted an interesting study which examined the association between phthalates exposure and serum uric acid in CD patients and explored the role of oxidative stress too. However, several parts require clarification, additional details, or methodological refinement to strengthen the manuscript before getting it published online.

Comments/Questions: -

- In the abstract, please clarify what "1.36-fold” by mentioning the unit

- include the percentage of the total effects mediated by oxidative stress in the abstract

- The methods and results are overly detailed. The authors must consider cutting off the section and including additional details as supplementary. also consider including table 3 and table 4 in supplementary file.

- I noticed that the patient characteristics you reported that all patients had normal kidney function (eGFR> 90) and nobody in the cohort was obese (BMI > 30), that needs to be noted as a limitation when discussing generalizability

- Including a more detailed explanation of figure legends would improve readability.

- I can’t see the ethical approval of this study in the supplementary file? Can you provide

- The sample size is small, and the sex imbalance is severe (88/29). The finding of no positive association in females is likely a due to the underpowered sample size, discuss this and acknowledge in the limitations

- -also consider adding a sentence about the call of future research to further validate the results

- get the manuscript reviewed by a native English speaker to ensure the clarity and readability of your MS with more grammatically correct English and punctuation.

**Do you want your identity to be public for this peer review?** For information about this choice, including consent withdrawal, please see our Privacy Policy

Reviewer #1: No

Reviewer #2: **Yes:** Abdallfatah Abdallfatah

---

## [Author Response · Author response to Decision Letter 1]

19 Jan 2026

Reviewer #1

1) Major concerns: With regards to the experimental design, there is a strong imbalance between male and female patients. Furthermore, the numbers of patients with hyperuricemia are too small, which limit the detection power of the method. Among the statistical methods, only the association analysis was applied, including Spearman correction, the generalized linear models. There is no comparison between those patients with hyperuricemia and those without it. Additionally, they use 10 urinary phthalates metabolites as a proxy for the phthalates exposure, which could be distorted by the metabolite functions of the patients. Why did not measure the levels in the serum since it has been used for uric acid measurement?

We sincerely thank you for your insightful and constructive comments regarding the trial design and methodological approach of our study. We acknowledge the limitations raised and have undertaken comprehensive analyses to address these concerns and strengthen the manuscript accordingly. Our point-by-point responses are detailed below.

We recognize the your concern regarding the imbalance in the gender ratio of our cohort. We agree that this is an important limitation. This imbalance was an inherent characteristic of our single center, non-selective, cross-sectional sampling strategy, and practical constraints prevented us from extending the study period to achieve a balanced recruitment. To directly mitigate the potential bias introduced by this imbalance and to explore potential sex-specific effects, we have performed subgroup analyses. The relatively small number of female samples did indeed significantly weaken the strength of the conclusion about female. Exploring the differences based on gender may be defined as an important direction for future research, which requires larger-scale, specially designed cohorts.

Regarding the concern about the small number of participants with hyperuricemia (HUA) and its impact on statistical power, we agree that the limited HUA sample size reduces the power for analyses using HUA as a outcome. We want to clarify that the primary outcome was the serum uric acid level as a continuous variable. This approach utilizes the full range of uric acid measurements across all participants, which maximizes statistical power and is not adversely affected by the small HUA subgroup. Therefore, our primary conclusions remain robust.

In response to the valuable suggestion to compare participants with and without HUA, both correlation analysis and regression analysis (Figure 2 and Table 2) have demonstrated a positive correlation between hyperuricemia and mPAEs. Given the large number of charts in the manuscript, we did not present the results of the group comparison.

Upon entering the human body, phthalates are rapidly hydrolyzed by metabolic enzymes such as esterases into their monoester metabolites. These metabolites are further transformed into more polar compounds, which are ultimately excreted in urine . Due to the short half-life of parent phthalates in the body, their concentrations in blood or urine are generally very low, making accurate detection challenging . In contrast, the metabolites remain in the body for a relatively longer period and occur at higher concentrations, making them more reliable biomarkers for reflecting recent exposure. Concerning the choice of biological sample for measuring phthalate exposure, your suggestion to use serum is valuable and noted. When initially designing this study, we referred to previous related research and found that almost all of them used urine sample. While serum is suitable for measuring uric acid, its complex composition can interfere with the quantification of phthalates. The use of urine for phthalate assessment is therefore methodologically sound and consistent with prevailing practices in environmental epidemiology. In the Method section, the reasons for utilizing urine samples have been provided.

We believe that the additional analyses and clarifications provided have significantly strengthened the manuscript. We are grateful for your comments, which have undoubtedly improved the quality of our work.

2) The tables and figures presented the same results. There are too many redundant data.

We thank you for this insightful observation. We agree that minimizing data redundancy is crucial for clarity.

Concerning Tables 1 and 2: We acknowledge that Table 1 (presenting absolute values of pollutant exposure) and Table 2 (showing the results of linear regression analyses) contain some overlapping data. However, we believe it is necessary to present both the descriptive statistics of the exposure levels and the subsequent statistical associations. Table 1 provides the foundational exposure metrics for the readers, while Table 2 quantifies the relationship between these exposures and the outcomes. We feel that both perspectives are important for a complete understanding of the study.

Action Taken on Tables 3 and 4: To address the concern of redundancy directly, we have moved Table 4 (which explored the mediating role of a different oxidative stress marker) to the Supplementary Materials. This leaves Table 3 in the main text.

Action Taken on Figures 3 and 4: Regarding the figures, we agree that Figure 4 served as supplementary information for Figure 3. Consequently, we have merged these two figures into a single, more comprehensive figure in the revised manuscript to eliminate repetition.

Supplementary Materials: The figures and tables in the Supplementary Materials primarily relate to the quality control of pollutant detection and more detailed results from the BKMR analysis. We believe these do not present significant redundancy with the main text content.

We hope that these revisions and explanations adequately address your concerns. We are grateful for your constructive suggestions, which have undoubtedly helped improve our manuscript. Should you have any further feedback, we would be pleased to make additional revisions.

3) The writings should be improved. There are many expression ambiguity and errors. For example, Lines 248-249 include redundant words and letters. Lines 103-104 include two acronyms for mPAEs while the first one seems an awkward appearance. Line 218, the statement “MEOHP was identified to be the major contributor to SUA levels” is too strong. The true meaning is the strongest association. The data did not lead to the conclusion of ‘major contributor’. “P-vale” in many places. The few sentences in the first paragraph of Discussion repeated themself.

Thank you very much for your valuable comments and suggestions concerning our manuscript. We sincerely appreciate the time and effort you have dedicated to reviewing our work. We agree with your assessment and have carefully revised the manuscript to address the issues you raised. Specifically:

Regarding the English writing: We apologize for the language errors in the original version. The manuscript has been thoroughly checked to improve clarity and correctness. The redundant acronym for mPAEs has been removed. All instances of “P-vale” have been corrected to “P-value” throughout the manuscript.

Concerning the strong statement: The sentence in Line 218, “MEOHP was identified to be the major contributor to SUA levels,” has been rephrased to more accurately reflect the findings, such as “MEOHP showed the strongest association with SUA levels”

On the repetitive sentences: The redundant expressions in the first paragraph of the Discussion section have been revised to improve conciseness and logical flow.

We believe that these revisions have significantly improved the quality of our manuscript. Thank you again for your insightful comments.

Reviewer #2

1) In the abstract, please clarify what "1.36-fold” by mentioning the unit.

Thank you for this valuable comment. We have revised the text in the abstract to clarify that the "1.36-fold" increase refers to the change of serum uric acid (SUA) concentration. The sentence now reads: fter BKMR analysis and multivariate adjustments, we found that the average SUA (μmol/L) would increase by 1.36-fold.

2) Include the percentage of the total effects mediated by oxidative stress in the abstract.

Thank you for this request. We have revised the text in the abstract: Furthermore, oxidative stress mediated approximately 5% of the association via two pathways, indicating it is a partial, but not primary, mechanism in this process. And correspondingly, we have made the same explanations in the results section.

3) The methods and results are overly detailed. The authors must consider cutting off the section and including additional details as supplementary. also consider including table 3 and table 4 in supplementary file.

Thank you for this constructive suggestion. We agree that moving some overly detailed content to the supplementary material will enhance the readability and focus of the main text. Following your advice, we have: Streamlined the Methods sections by removing overly detailed descriptions. Relocated Table 3 and 4 to the supplementary file. These changes have been made in the revised manuscript, and the relocated/content is clearly indicated in the supplementary file. Through these measures, we have reduced the word count of the main text by approximately 500 words. We believe the manuscript is now more concise and accessible to readers.

4) I noticed that the patient characteristics you reported that all patients had normal kidney function (eGFR> 90) and nobody in the cohort was obese (BMI > 30), that needs to be noted as a limitation when discussing generalizability.

Thank you for this insightful comment. We completely agree that the homogeneity of our study population in terms of normal renal function and the absence of obesity may limit the generalizability of our findings to the broader CD population. We have now explicitly acknowledged this as a limitation in the Discussion section. The added text reads: "Furthermore, our study cohort consisted exclusively of non-obese CD patients with normal renal function. While this homogeneity strengthens the internal validity of our findings by minimizing confounding from these strong determinants of uric acid levels, it may limit the generalizability of our results to CD patients with obesity or renal impairment."

5) Including a more detailed explanation of figure legends would improve readability.

Thank for your valuable suggestion. We agree that comprehensive figure legends are essential for reader comprehension. In accordance with this comment, we have thoroughly revised all figure legends to provide more detailed explanations. The legends now include clearer descriptions of the experimental groups, statistical methods depicted (e.g., BKMR analysis), and the meaning of key symbols or abbreviations used directly in the figures. We believe these enhancements significantly improve the clarity and self-contained nature of the figures, allowing readers to grasp the key findings more readily.

6) I can’t see the ethical approval of this study in the supplementary file? Can you provide

Thank you for your thoughtful review and for raising this important point regarding the ethical approval for our study. We confirm that the ethical approval document was submitted as part of the supplementary materials during the initial manuscript submission, in accordance with the journal’s guidelines. For your convenience, we have now included both the original ethical approval certificate in Chinese and its corresponding English translation as images attached to this response letter (Please see Attachment below). We appreciate your time and consideration. Please let us know if you require any additional information or clarifications.

7 ) The sample size is small, and the sex imbalance is severe (88/29). The finding of no positive association in females is likely a due to the underpowered sample size, discuss this and acknowledge in the limitations.

8) also consider adding a sentence about the call of future research to further validate the results

Thank for your detail suggestion. We agree that pointing out directions for future research would strengthen the impact and completeness of our paper. In response to your comment, we have added a sentence in the Conclusion and Limitations section to address this.

9) get the manuscript reviewed by a native English speaker to ensure the clarity and readability of your MS with more grammatically correct English and punctuation.

Thank you for the valuable feedback regarding language clarity. We acknowledge the importance of impeccable English for scientific communication. To address this thoroughly, the manuscript has undergone an extensive round of language revision. Additionally, we utilized grammar checking tools (Grammarly) and had the manuscript proofread by several colleagues fluent in academic English. We believe these efforts have substantially improved the readability of the manuscript, and we sincerely hope it now meets the journal’s high standards.

---

## [Decision Letter · Decision Letter 1]

2 Feb 2026

Phthalates exposure and serum uric acid level in patients with Crohn's disease: a cross-sectional study

PONE-D-25-46092R1

Dear Liu,

We’re pleased to inform you that your manuscript has been judged scientifically suitable for publication and will be formally accepted for publication once it meets all outstanding technical requirements.

Kind regards,

Saheed Abiola Raheem, PhD

Academic Editor

PLOS One

Additional Editor Comments (optional):

I commend the Authors for the painstaking efforts in addressing the concerns and comments suggested by the reviewers. We appreciate your efforts, cooperation, and understanding.

Reviewers' comments:

Reviewer's Responses to Questions

**Comments to the Author**

Reviewer #1: All comments have been addressed

Reviewer #2: All comments have been addressed

2. Is the manuscript technically sound, and do the data support the conclusions?

Reviewer #1: Yes

Reviewer #2: Yes

3. Has the statistical analysis been performed appropriately and rigorously?

Reviewer #1: Yes

Reviewer #2: Yes

4. Have the authors made all data underlying the findings in their manuscript fully available?

Reviewer #1: Yes

Reviewer #2: Yes

5. Is the manuscript presented in an intelligible fashion and written in standard English?

Reviewer #1: Yes

Reviewer #2: Yes

Reviewer #1: The reviewers' comments have been addressed adequately. I have no further concerns.

Reviewer #2: I have no more comments for authors, very well done job, Thanks for addressing my comments and congratulations

**Do you want your identity to be public for this peer review?** For information about this choice, including consent withdrawal, please see our Privacy Policy

Reviewer #1: No

Reviewer #2: **Yes:** Abdallfatah

---

## [Editor Report · Acceptance letter]

PONE-D-25-46092R1

PLOS One

Dear Dr. Liu,

I'm pleased to inform you that your manuscript has been deemed suitable for publication in PLOS One. Congratulations! Your manuscript is now being handed over to our production team.

Kind regards,

on behalf of

Dr. Saheed Abiola Raheem

Academic Editor

PLOS One